# BOOTSTRAPPING PARALLEL ANCHORS FOR RELATIVE REPRESENTATIONS

**Irene Cannistraci**[1]   **Luca Moschella**[1]   **Valentino Maiorca**[1]

**Marco Fumero**[1]   **Antonio Norelli**[1]   **Emanuele Rodolà**[1]

[1]Sapienza University of Rome

## ABSTRACT

The use of relative representations for latent embeddings has shown potential in enabling latent space communication and zero-shot model stitching across a wide range of applications. Nevertheless, relative representations rely on a certain amount of parallel anchors to be given as input, which can be impractical to obtain in certain scenarios. To overcome this limitation, we propose an optimization-based method to discover new parallel anchors from a limited known set (*seed*). Our approach can be used to find semantic correspondence between different domains, align their relative spaces, and achieve competitive results in several tasks.

## 1 INTRODUCTION

Over the past few years, several studies have acknowledged how successful neural networks typically learn comparable representations regardless of their architecture, task, or domain (Li et al., 2016; Kornblith et al., 2019; Vulić et al., 2020). In line with this trend, Moschella et al. (2023) introduced the concept of relative representation, aiming to generate comparable latent spaces and enable zero-shot stitching to handle new, unseen tasks without requiring additional training. The approach consists in representing each data sample through latent similarities with respect to a set of training samples, denoted as *anchors*. This procedure transforms the absolute reference frame to a relative coordinate system defined by the anchors. To enable tasks like multimodal learning, this approach requires a semantic connection between the anchors of two data domains, denoted as *parallel anchors*. This correspondence, which must be provided as input, allows domain comparison and links their respective latent spaces (Norelli et al., 2022). However, obtaining a sufficient number of parallel anchors in specific applications can be challenging or impossible, hindering the use of relative representations. We focus on the scenario where there are only a very limited number of parallel anchors available, called *seed*, and we aim to expand this initial set through an Anchor Optimization (AO) process. Our method achieves competitive performance in NLP and Vision domains while significantly reducing the number of required parallel anchors by *one order of magnitude*.

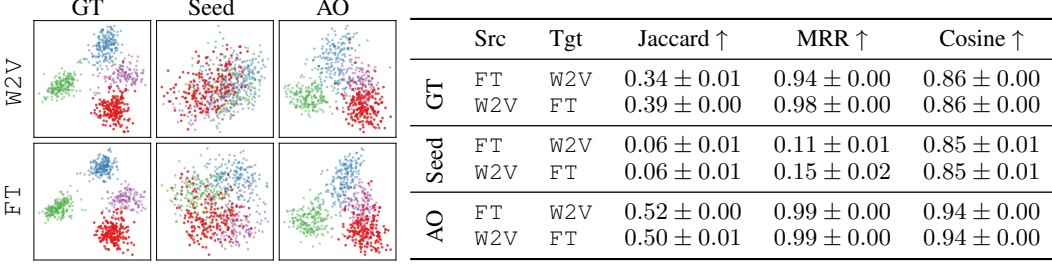

| | Src | Tgt | Jaccard ↑ | MRR ↑ | Cosine ↑ |
|---|---|---|---|---|---|
| **GT** | FT | W2V | $0.34 \pm 0.01$ | $0.94 \pm 0.00$ | $0.86 \pm 0.00$ |
| | W2V | FT | $0.39 \pm 0.00$ | $0.98 \pm 0.00$ | $0.86 \pm 0.00$ |
| **Seed** | FT | W2V | $0.06 \pm 0.01$ | $0.11 \pm 0.01$ | $0.85 \pm 0.01$ |
| | W2V | FT | $0.06 \pm 0.01$ | $0.15 \pm 0.02$ | $0.85 \pm 0.01$ |
| **AO** | FT | W2V | $0.52 \pm 0.00$ | $0.99 \pm 0.00$ | $0.94 \pm 0.00$ |
| | W2V | FT | $0.50 \pm 0.01$ | $0.99 \pm 0.00$ | $0.94 \pm 0.00$ |

Table 1: Qualitative (*left*) and quantitative (*right*) evaluation of the AO method in the retrieval task.

## 2 METHOD

Let us be given two domains $\mathcal{X}$ and $\mathcal{Y}$ and corresponding learned embedding functions $E_{\mathcal{X}} : \mathcal{X} \to \mathbb{R}^n$ and $E_{\mathcal{Y}} : \mathcal{Y} \to \mathbb{R}^m$, where possibly $n \neq m$. Given two sets of anchors $\mathcal{A}_{\mathcal{X}} \subset \mathcal{X}$ and

$\mathcal{A}_{\mathcal{Y}} \subset \mathcal{Y}$, we define *parallel anchors* a subset of pairs $\mathcal{A}_p \subseteq \mathcal{A}_{\mathcal{X}} \times \mathcal{A}_{\mathcal{Y}}$ in semantic correspondence, e.g., images and captions as in Norelli et al. (2022). The relative representations for a sample $x \in \mathcal{X}$ (same for $\mathcal{Y}$) is computed as follows: $rr(x, \mathcal{A}_{\mathcal{X}}) = E_{\mathcal{X}}(x)\mathbf{A}_{\mathcal{X}}^T$, where $\mathbf{A}_{\mathcal{X}} = \bigoplus_{a \in \mathcal{A}_{\mathcal{X}}} E_{\mathcal{X}}(a)$, and $\bigoplus$ denotes the row-wise concatenation operator. We assume all embeddings are rescaled to unit norm, i.e., $\forall x\ ||E(x)|| = 1$. This corresponds to the choice of cosine similarity as a similarity function, according to the setting of Moschella et al. (2023).

In this work, we introduce an optimization procedure that reduces the required number of parallel anchors by one order of magnitude. Our method does not require complete knowledge of $\mathcal{A}_p$ but only of few initial *seed* anchors, denoted as $\mathcal{L} = \mathcal{L}_{\mathcal{X}} \times \mathcal{L}_{\mathcal{Y}} \subseteq \mathcal{A}p$, where $|\mathcal{L}| \ll |\mathcal{A}p|$. With no prior knowledge of $\mathcal{A}_{\mathcal{Y}}$, we initialize the optimization process by approximating $\mathbf{A}_{\mathcal{Y}} \approx \widetilde{\mathbf{A}}_{\mathcal{Y}}$ with the known seed $\mathbf{A}_{\mathcal{L}_{\mathcal{Y}}} = \bigoplus_{a \in \mathcal{L}_{\mathcal{Y}}} E_{\mathcal{Y}}(a)$ concatenated with $|\mathcal{A}_p| - |\mathcal{L}|$ random embeddings, i.e. $\widetilde{\mathbf{A}}_{\mathcal{Y}} = \mathbf{A}_{\mathcal{L}_{\mathcal{Y}}} \oplus \mathbf{N}$, with $\mathbf{N} \sim \mathcal{N}(0, \mathbf{I})$. We define the following objective function optimizing over $\widetilde{\mathbf{A}}_{\mathcal{Y}}$:

$$\underset{\widetilde{\mathbf{A}}_{\mathcal{Y}} \text{ s.t. } ||a||_2 = 1\ \forall a \in \widetilde{\mathbf{A}}_{\mathcal{Y}}}{\arg\min} \sum_{y \in \mathcal{Y}} MSE(rr(\Pi(y), \mathcal{A}_{\mathcal{X}}), E_{\mathcal{Y}}(y)\widetilde{\mathbf{A}}_{\mathcal{Y}}^T) \tag{1}$$

where $\Pi : \mathcal{Y} \to \mathcal{X}$ is a correspondence estimated at each optimization step by the Sinkhorn (Cuturi, 2013) algorithm exploiting the initial seed and the current approximation of the remaining anchors: $\Pi = sinkhorn_{(x,y) \in \mathcal{X} \times \mathcal{Y}}(rr(x, \mathcal{A}_{\mathcal{X}}), E_{\mathcal{Y}}(y)\widetilde{\mathbf{A}}_{\mathcal{Y}}^T)$. After convergence, $\widetilde{\mathbf{A}}_{\mathcal{Y}}$ is discretized into $\widetilde{\mathcal{A}}_{\mathcal{Y}} \subseteq \mathcal{Y}$ considering the nearest embeddings in $E_{\mathcal{Y}}(\mathcal{Y})$. Further details in Appendix A.2.

## 3 EXPERIMENTS

This section assesses the effectiveness of our AO method. We utilize 15 anchor to approximate 300 parallel anchors that serve as ground truth in all downstream tasks. Specifically, we compare the performance of our method against two different baselines: (1) *GT*, the Ground Truth employs all the anchors that our method aims to semantically approximate, (2) *Seed*, exploits only the seed anchors. For more information on the implementation please refer to the Appendix A.2 and the code[1].

Our method effectively discovers parallel anchors in the NLP and Vision domains, as demonstrated in Tables 1, 4, 5 and 7. Specifically, we explore different word embeddings and pre-trained foundational visual encoders, and assess the quality of the discovered anchors through a retrieval task. Results demonstrate that, when given the same number of starting anchors, our method outperforms the approach that relies solely on those without optimizing. Moreover, our results are *comparable or superior* to those obtained with all the ground truth parallel anchors. Furthermore, Table 6 demonstrates that our method can discover parallel anchors across different domains: the method finds aligned Amazon reviews in different languages with unavailable ground truth. Using only 15 *OOD* (Moschella et al., 2023) parallel anchors, our method enables zero-shot stitching, allowing to train a classifier on one language and perform predictions on another one.

Table 2: Cross-lingual zero-shot stitching performance evaluation.

| Dec. | Enc. | GT | | Seed | | AO | |
| | | Fscore | MAE | Fscore | MAE | Fscore | MAE |
| --- | --- | --- | --- | --- | --- | --- | --- |
| en | es | $0.51 \pm 0.01$ | $0.67 \pm 0.02$ | $0.44 \pm 0.01$ | $0.80 \pm 0.01$ | $0.48 \pm 0.01$ | $0.70 \pm 0.02$ |
| es | en | $0.50 \pm 0.02$ | $0.72 \pm 0.04$ | $0.41 \pm 0.01$ | $0.92 \pm 0.02$ | $0.46 \pm 0.01$ | $0.76 \pm 0.02$ |

## 4 CONCLUSIONS, FUTURE WORKS, AND LIMITATIONS

In this paper, we presented a novel method to compute robust relative representations even in scenarios where only a reduced number of parallel anchors is available. The method expands semantic correspondence between data domains without prior knowledge and achieves comparable results with *one order of magnitude fewer* parallel anchors. This approach has notable implications for latent space communication across domains with limited knowledge about semantic correspondence. Future research is needed to remove the need for an initial parallel seed.

---

[1]Fully reproducible codebase at: `https://github.com/icannistraci/bootstrapping-ao`

ACKNOWLEDGEMENTS

This work is supported by the ERC Grant no.802554 "SPECGEO" and PRIN 2020 project no.2020TA3K9N "LEGO.AI".

URM STATEMENT

The first author meets the URM criteria of ICLR 2023 Tiny Papers Track.

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

## A  APPENDIX

### A.1  RELATED WORKS

In recent years, numerous studies (Lenc & Vedaldi, 2015; Mikolov et al., 2013b; Li et al., 2016; Lample et al., 2018; Morcos et al., 2018; Tsitsulin et al., 2020; Kornblith et al., 2019; Vulić et al., 2020; Antonello et al., 2021; Bonheme & Grzes, 2022; Barannikov et al., 2022; Norelli et al., 2022) have recognized that neural networks tend to learn comparable representations regardless of their architecture, task, or domain when trained on semantically similar data. This observation can be exploited to enable various applications, such as model stitching (Lenc & Vedaldi, 2015; Bansal et al., 2021; Csiszárik et al., 2021; Gygli et al., 2021; Biondi et al., 2021; Bianchi et al., 2020), latent model comparison or supervision and more. In particular, Moschella et al. (2023) introduced the framework of relative representation, which aims to unify the representations learned from semantically similar data. Relative representations have demonstrated potential in facilitating communication within latent embeddings and enabling zero-shot stitching across various applications, relying on parallel anchors to link different domains. Our work aims to minimize the explicit supervision required for latent communication by reducing the reliance on parallel anchors to the minimum necessary and automatically expand the provided semantic correspondence between domains.

### A.2  IMPLEMENTATION DETAILS

This section provides further details about the optimization procedure and the experiments.

**Optimization Method**   Algorithm 1 outlines the pseudocode for the optimization procedure described in Section 2, while Table 3 details the hyperparameters. The method initializes $\widetilde{\mathbf{A}}_{\mathcal{Y}}$ and optimizes it iteratively. At each step, the Sinkhorn algorithm computes a rough estimate of the permutation between the two relative spaces. The loss function minimized in our optimization procedure is the MSE, with particular emphasis placed on ensuring that the optimized parameters $\widetilde{\mathbf{A}}_{\mathcal{Y}}$ adhere to unit norm using Casado (2019). This not only ensures the effectiveness of the optimization but also reduces the search space.

---

**Algorithm 1** Anchor Optimization

---

1: Initialize $\widetilde{\mathbf{A}}_{\mathcal{Y}} = \mathbf{A}_{\mathcal{L}_{\mathcal{Y}}} \oplus \mathbf{N}$, with $\mathbf{N} \in \mathcal{N}(0, \mathbf{I})$ and $|\mathbf{N}| = |\mathcal{A}_p| - |\mathcal{L}|$
2: Compute the relative representations of samples in $\mathcal{X}$ as $\mathbf{R}_{\mathcal{X}} = \bigoplus_{x \in \mathcal{X}} rr(x, \mathcal{A}_{\mathcal{X}})$
3: **for** $K$ steps **do**
4:     Compute the relative representations of samples in $\mathcal{Y}$ as $\mathbf{R}_{\mathcal{Y}} = \bigoplus_{y \in \mathcal{Y}} E_{\mathcal{Y}}(y)\widetilde{\mathbf{A}}_{\mathcal{Y}}^{T}$
5:     Estimate the permutation between $\mathbf{R}_{\mathcal{Y}}$ and $\mathbf{R}_{\mathcal{X}}$ with $\Pi = sinkhorn(\mathbf{R}_{\mathcal{X}}, \mathbf{R}_{\mathcal{Y}})$
6:     Permute $\mathbf{R}_{\mathcal{Y}}$ according to $\Pi$
7:     Compute the error $MSE(\mathbf{R}_{\mathcal{X}}, \mathbf{R}_{\mathcal{Y}})$
8:     Optimize $\widetilde{\mathbf{A}}_{\mathcal{Y}}$ to minimize the error, while abiding to the constraint $||a||_2 = 1 \ \forall a \in \widetilde{\mathbf{A}}_{\mathcal{Y}}$
9: **end for**
10: **return** the nearest neighbours of $\widetilde{\mathbf{A}}_{\mathcal{Y}}$ in $E_{\mathcal{Y}}(\mathcal{Y})$

---

**Retrieval Task**   We choose two English word embeddings trained on different data but with a partially shared vocabulary from which we extract $\approx$ 20K words: `FastText` (Bojanowski et al., 2017) and `Word2Vec` (Mikolov et al., 2013a). For testing the AO method, we select 15 seed anchors and shuffle the two embedding spaces to break their correspondence. Then, we choose 285 additional random anchors for one of the spaces while we use our optimization method to discover the associated 285 parallel anchors in the other one. Next, the absolute embeddings of each space are converted to their relative representations using the 300 optimized parallel anchors. For each word $w$, we consider its corresponding encodings $x$ and $y$ in the source and target space and validate their quality through a retrieval task. To facilitate a comparison with the relative representation baseline (Moschella et al., 2023), we employ the same evaluation metrics: (i) *Jaccard*: the discrete Jaccard similarity between the set of word neighbors of $x$ in source and target; (ii) *Mean Reciprocal Rank (MRR)*: measures the (reciprocal) ranking of $w$ among the top-k neighbors of $x$ in the target space; (iii) *Cosine*: measures the cosine similarity between $x$ and $y$. Results for the *GT* and *seed* methods

Table 3: Hyperparameter for the AO method in *retrieval* and *zero-shot stitching* tasks.

| Hyperparameter | Retrieval | Zero-shot stitching |
|---|---|---|
| Random seed | 0, 1, 2, 3, 4 | 0, 1, 2, 3, 4 |
| Number of anchors to approximate | 300 | 300 |
| Number of seed anchors | 15 | 15 |
| Number of optimization steps | 250 | 125 |
| Learning Rate | 0.02 | 0.05 |
| Optimizer | Adam | Adam |
| Loss | MSE | MSE |
| Sinkhorn eps | 1e-4 | 1e-4 |
| Sinkhorn stop error | 1e-5 | 1e-5 |
| Number of Sinkhorn steps | 1 | 1 |

are obtained by using all the given 300 anchors that our method aims to semantically approximate and only the 15 seed anchors, respectively.

| | Source | Target | Jaccard ↑ | MRR ↑ | Cosine ↑ |
|---|---|---|---|---|---|
| GT | FT | FT | $1.00 \pm 0.00$ | $1.00 \pm 0.00$ | $1.00 \pm 0.00$ |
| | | W2V | $0.34 \pm 0.01$ | $0.94 \pm 0.00$ | $0.86 \pm 0.00$ |
| | W2V | FT | $0.39 \pm 0.00$ | $0.98 \pm 0.00$ | $0.86 \pm 0.00$ |
| | | W2V | $1.00 \pm 0.00$ | $1.00 \pm 0.00$ | $1.00 \pm 0.00$ |
| Seed | FT | FT | $1.00 \pm 0.00$ | $1.00 \pm 0.00$ | $1.00 \pm 0.00$ |
| | | W2V | $0.06 \pm 0.01$ | $0.11 \pm 0.01$ | $0.85 \pm 0.01$ |
| | W2V | FT | $0.06 \pm 0.01$ | $0.15 \pm 0.02$ | $0.85 \pm 0.01$ |
| | | W2V | $1.00 \pm 0.00$ | $1.00 \pm 0.00$ | $1.00 \pm 0.00$ |
| AO | FT | FT | $1.00 \pm 0.00$ | $1.00 \pm 0.00$ | $1.00 \pm 0.00$ |
| | | W2V | $0.52 \pm 0.00$ | $0.99 \pm 0.00$ | $0.94 \pm 0.00$ |
| | W2V | FT | $0.50 \pm 0.01$ | $0.99 \pm 0.00$ | $0.94 \pm 0.00$ |
| | | W2V | $1.00 \pm 0.00$ | $1.00 \pm 0.00$ | $1.00 \pm 0.00$ |

Table 4: Complete results for the selected experiments are reported in Table 1. Qualitative (*left*) and quantitative (*right*) comparisons of the three methods when optimizing the Word2Vec space. All metrics are calculated with $K = 10$ averaged over 20k words across five random seeds.

**Zero-shot stitching task** We investigate the *Cross-lingual* text classification task on the multi-lingual Amazon Reviews dataset (Keung et al., 2020) to demonstrate a practical application of our method. We use the fine-grained formulation of the task, where the goal is to predict the star rating given a review (i.e., five classes to predict) and measure performance using FScore and Mean Absolute Error (MAE) metrics. To evaluate the effectiveness of our method, we utilize two different pre-trained language-specific RoBERTa transformers (Liu et al., 2019) and test their *zero-shot stitching performance on languages that were not seen during training*. Specifically, we evaluate our method using English and Spanish languages with PlanTL-GOB-ES/roberta-base-bne and roberta-base, respectively. Similar to the implementation details in word embeddings discussed in Section A.2, we begin by choosing 15 random parallel anchors as seed and then select an additional 285 random anchors for the Spanish space. We then apply our optimization method to discover the remaining 285 parallel anchors for the English space. Next, the absolute embeddings of each space are converted to relative representations using the 300 optimized parallel anchors. Table 6 presents the *Cross-lingual* zero-shot stitching performance of our approach, demonstrating its efficacy in learning to solve a downstream task on a specific language or transformer and making accurate predictions while relying on the discovered anchors.

**Tools & Technologies** We use the following tools in all the experiments presented in this work:

- *PyTorch Lightning*, to ensure reproducible results while also getting a clean and modular codebase;

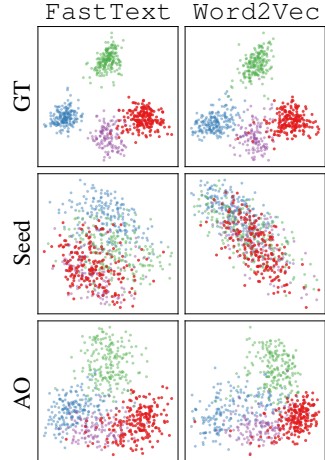

| | | Source | Target | Jaccard ↑ | MRR ↑ | Cosine ↑ |
|---|---|---|---|---|---|---|
| | GT | FT | FT | $1.00 \pm 0.00$ | $1.00 \pm 0.00$ | $1.00 \pm 0.00$ |
| | | | W2V | $0.34 \pm 0.01$ | $0.94 \pm 0.00$ | $0.86 \pm 0.00$ |
| | | W2V | FT | $0.39 \pm 0.00$ | $0.98 \pm 0.00$ | $0.86 \pm 0.00$ |
| | | | W2V | $1.00 \pm 0.00$ | $1.00 \pm 0.00$ | $1.00 \pm 0.00$ |
| | Seed | FT | FT | $1.00 \pm 0.00$ | $1.00 \pm 0.00$ | $1.00 \pm 0.00$ |
| | | | W2V | $0.06 \pm 0.01$ | $0.11 \pm 0.01$ | $0.85 \pm 0.01$ |
| | | W2V | FT | $0.06 \pm 0.01$ | $0.15 \pm 0.02$ | $0.85 \pm 0.01$ |
| | | | W2V | $1.00 \pm 0.00$ | $1.00 \pm 0.00$ | $1.00 \pm 0.00$ |
| | AO | FT | FT | $1.00 \pm 0.00$ | $1.00 \pm 0.00$ | $1.00 \pm 0.00$ |
| | | | W2V | $0.49 \pm 0.00$ | $0.98 \pm 0.00$ | $0.93 \pm 0.00$ |
| | | W2V | FT | $0.50 \pm 0.00$ | $0.99 \pm 0.00$ | $0.93 \pm 0.00$ |
| | | | W2V | $1.00 \pm 0.00$ | $1.00 \pm 0.00$ | $1.00 \pm 0.00$ |

Table 5: Corresponding results to those reported in Table 4, illustrating the performance of the model when optimizing the anchors in the other latent space. Qualitative (*left*) and quantitative (*right*) comparisons of the three methods when optimizing the `FastText` space. All metrics are calculated with $K = 10$ averaged over 20k words across five random seeds.

Table 6: Complete results for the selected experiments are reported in Table 2. Cross-lingual zero-shot stitching performance evaluation. The table reports the mean weighted F1 and MAE on Amazon Reviews fine-grained dataset across five random seeds.

| Decoder | Encoder | GT | | Seed | | AO | |
|---|---|---|---|---|---|---|---|
| | | Fscore | MAE | Fscore | MAE | Fscore | MAE |
| en | en | $0.64 \pm 0.01$ | $0.43 \pm 0.01$ | $0.50 \pm 0.01$ | $0.69 \pm 0.01$ | $0.62 \pm 0.01$ | $0.44 \pm 0.01$ |
| | es | $0.51 \pm 0.01$ | $0.67 \pm 0.02$ | $0.44 \pm 0.01$ | $0.80 \pm 0.01$ | $0.48 \pm 0.01$ | $0.70 \pm 0.02$ |
| es | en | $0.50 \pm 0.02$ | $0.72 \pm 0.04$ | $0.41 \pm 0.01$ | $0.92 \pm 0.02$ | $0.46 \pm 0.01$ | $0.76 \pm 0.02$ |
| | es | $0.60 \pm 0.01$ | $0.45 \pm 0.01$ | $0.48 \pm 0.01$ | $0.70 \pm 0.01$ | $0.61 \pm 0.01$ | $0.44 \pm 0.01$ |

- *GeoTorch* Casado (2019), to constrain optimized anchor vectors to have unit norm;
- *Fast, Memory-Efficient Approximate Wasserstein Distances*, to optimize anchor vectors;
- *Transformers by HuggingFace*, to get ready-to-use transformers for both text and images;
- *Datasets by HuggingFace*, to access most of the NLP datasets and CIFAR10 (Krizhevsky et al., 2009) for CV;
- *DVC* (Kuprieiev et al., 2022), for data versioning;

### A.3 ADDITIONAL EXPERIMENTS

Building upon the methodology presented in the word embeddings experiment introduced in Section 3 and detailed in Appendix A.2, we generalize the retrieval results from the NLP to the Vision domain. To achieve this, we first extract $\approx 20K$ images from `CIFAR-10`. We then encode these images using two variants of the `VIT` transformer model: the `VIT_base_patch16` model, which is pre-trained on `JFT-300M` (Sun et al., 2017) and `ImageNet` (Deng et al., 2009), and the `VIT_small_patch16` model, which is pre-trained solely on `ImageNet`. The two models have respective encoding dimensions of 768 and 384. We follow the same experimental setting, comparing our model against two different baselines (*GT* and *Seed* methods), and we evaluate the performance with *Jaccard*, *MRR* and *Cosine* metrics. Results are reported in Table 7.

Table 7: Generalization of the results described in Section 3, from word embeddings to images using the CIFAR-10 dataset. The table reports the mean results for each metric and its standard deviation across five different random seeds.

| Mode | Type | Source | Target | Jaccard ↑ | MRR ↑ | Cosine ↑ |
|------|------|--------|--------|-----------|-------|----------|
| GT | Absolute | ViT-base | ViT-base | $1.00 \pm 0.00$ | $1.00 \pm 0.00$ | $1.00 \pm 0.00$ |
| | | | ViT-small | - | - | - |
| | | ViT-small | ViT-base | - | - | - |
| | | | ViT-small | $1.00 \pm 0.00$ | $1.00 \pm 0.00$ | $1.00 \pm 0.00$ |
| | Relative | ViT-base | ViT-base | $1.00 \pm 0.00$ | $1.00 \pm 0.00$ | $1.00 \pm 0.00$ |
| | | | ViT-small | $0.11 \pm 0.00$ | $0.27 \pm 0.01$ | $0.97 \pm 0.00$ |
| | | ViT-small | ViT-base | $0.10 \pm 0.00$ | $0.28 \pm 0.01$ | $0.97 \pm 0.00$ |
| | | | ViT-small | $1.00 \pm 0.00$ | $1.00 \pm 0.00$ | $1.00 \pm 0.00$ |
| Seed | Absolute | ViT-base | ViT-base | $1.00 \pm 0.00$ | $1.00 \pm 0.00$ | $1.00 \pm 0.00$ |
| | | | ViT-small | - | - | - |
| | | ViT-small | ViT-base | - | - | - |
| | | | ViT-small | $1.00 \pm 0.00$ | $1.00 \pm 0.00$ | $1.00 \pm 0.00$ |
| | Relative | ViT-base | ViT-base | $1.00 \pm 0.00$ | $1.00 \pm 0.00$ | $1.00 \pm 0.00$ |
| | | | ViT-small | $0.03 \pm 0.00$ | $0.03 \pm 0.01$ | $0.97 \pm 0.00$ |
| | | ViT-small | ViT-base | $0.03 \pm 0.00$ | $0.04 \pm 0.01$ | $0.96 \pm 0.00$ |
| | | | ViT-small | $1.00 \pm 0.00$ | $1.00 \pm 0.00$ | $1.00 \pm 0.00$ |
| AO | Absolute | ViT-base | ViT-base | $1.00 \pm 0.00$ | $1.00 \pm 0.00$ | $1.00 \pm 0.00$ |
| | | | ViT-small | - | - | - |
| | | ViT-small | ViT-base | - | - | - |
| | | | ViT-small | $1.00 \pm 0.00$ | $1.00 \pm 0.00$ | $1.00 \pm 0.00$ |
| | Relative | ViT-base | ViT-base | $1.00 \pm 0.00$ | $1.00 \pm 0.00$ | $1.00 \pm 0.00$ |
| | | | ViT-small | $0.10 \pm 0.01$ | $0.23 \pm 0.03$ | $0.97 \pm 0.00$ |
| | | ViT-small | ViT-base | $0.10 \pm 0.00$ | $0.28 \pm 0.01$ | $0.97 \pm 0.00$ |
| | | | ViT-small | $1.00 \pm 0.00$ | $1.00 \pm 0.00$ | $1.00 \pm 0.00$ |

