# OpenReview forum: "Bootstrapping Parallel Anchors for Relative Representations"
_ICLR.cc/2023/TinyPapers — Submitted to Tiny Papers @ ICLR 2023_

### Official Review · Reviewer_t3An · 2023-03-31

**Confidence:** 4

**Summary Of Contributions:**

The paper proposes a method for aligning/stitching the latent space of different domains using a small set of parallel anchors (called seeds). The approach involves alternately optimizing the latent permutation and aligned projection. The proposed method is effective and reduces the number of seeds needed for stitching different domains.

**Rating:**

High Potential (HP): a submission which meets the reviewing criteria and has potential to make an impact on the field

**Strengths And Weaknesses:**

### Strengths

1. Good research question. The problem of aligning the latent space of pretrained models using a few parallel samples is interesting and has potential applications in different domains. Reducing the parallel data points required for the alignment is thus important.

2. Effective method. The proposed method is backed up by quantitative results and visualizations that demonstrate its effectiveness.

3. Straightforward algorithm. The algorithm is designed without ad-hoc components, making it easy to follow.

### Weaknesses

1. Writing clarity. The writing could be polished to improve the clarity of the paper, especially considering that Tiny Papers should have a broader audience. The authors are encouraged to provide a better explanation of their methods in the main paper.

**Suggested Changes:**

1. Add a table of notions to help readers understand the terminologies used in the paper.

2. Expand on the explanation of the authors' ideas and methods in the main paper, including their motivation and design, while moving some of the experiments to the appendix.

---

### Official Review · Reviewer_qk6p · 2023-04-01

**Confidence:** 4

**Summary Of Contributions:**

Anchor optimization with limited seeds(limited parallel anchors) for relative representation: unfortunately no reproducible code provided

**Rating:**

Great Start (GS): a submission which meets some of the reviewing criteria but has room for improvement

**Strengths And Weaknesses:**


Strengths
1. Author introduces a Novel method to reduce no of parallel anchor for relative representation
2. Commendable to do both Qualitative & quantitative measures of evaluation
3. author suggests this approach(Anchor optimization) has competitive performance in NLP & CV domains
4. based on this paper, Anchor Optimization outperforms Ground Truth parallel anchors
5. Author has done a great job explaining algorithmic approach provided in appendix
6. Anchor Optimization hyperparameters in retrieval & zero-shot stitching tasks information provided in appendix
7. author has also provided the tools & technologies utilized in their approach (PyTorch Lightning, GeoTorch, Hugging face transformers & datasets, DVC,Fast, Memory-Efficient Approximate Wasserstein Distances)
Weakness
1. The author has done thorough research on this paper, explained the algorithmic approach in appendix, but failed to provide code to make these results reproducible.


**Suggested Changes:**

1. Do attempt to provide reproducible code, the evaluation criteria is highly based on this.
2. Though the authors have done great research easily making this top 20% of the tiny papers, unable to provide higher rating due to non-availability of reproducible code.

---

### Author Response · Authors · 2023-04-14
**General Response**

Dear Program Chairs, Area Chairs, and Reviewers,

Thank you for your feedback on our manuscript. We appreciate the valuable insights and suggestions for improvement that you have provided.

We want to reassure you that the code is available and the experiments are reproducible. Unfortunately, *it was not possible to submit the code as supplementary material*. We will soon release the code to ensure the reproducibility of our results for everyone.

Moreover, we will improve the manuscript's clarity with the suggested changes.

We appreciate the recommendation to present our work and look forward to sharing our research with the community!

Best regards

---

### Author Response · Authors · 2023-05-30
**Opt-in archival preference**

We confirm we would like to opt-in for archival on DBLP.

---

> ### Comment · Area_Chair_75AA · 2023-06-07
> **Check for archival**
>
> This work meets the threshold for archival, contents the URM statement and is deanonymized.

---

### Meta-Review · Area_Chair_75AA · 2023-04-07

**Recommendation:** Invite to present
**Confidence:** 4

**Metareview:**

This paper proposes a novel method, Anchor Optimization, for aligning and stitching the latent space of different domains using a limited set of parallel anchors. The proposed method involves alternately optimizing the latent permutation and aligned projection. While the method is effective and has competitive performance, the paper lacks reproducible code, and the writing could be polished for clarity.

This work is of pretty high quality. It studies an interesting research problem of aligning the latent space of different domains. Given properly motivated writing, it introduces an alternate optimization algorithm to reduce the number of anchors needed for the latent space alignment. Experiments are comprehensive and convincing. It clearly meets Clarity and Correctness.

However, the largest concern raised by reviewers is the reproducibility of this work. Authors are expected to release the code to ensure reproducibility.

**Summary:**

This paper proposes a novel method, Anchor Optimization, for aligning and stitching the latent space of different domains using a limited set of parallel anchors.  The proposed method is effective and has competitive performance.

**Reason For Not Giving A Higher Recommendation:**

The reviewers recognized the high quality of the work, including comprehensive experiments, clear and correct presentation of the method, and effective algorithm design. Therefore, the reviewers did not give a lower recommendation.

**Reason For Not Giving A Lower Recommendation:**

The reviewers suggest that the authors release the code to ensure the reproducibility of the results. Additionally, the writing clarity could be polished, including providing a better explanation of their methods in the main paper, and adding a table of notions to help readers understand the terminologies used in the paper.

---

### Decision · Program_Chairs · 2023-04-08

Invite to present